# Climate-mediated cooperation promotes niche expansion in burying beetles

**Syuan-Jyun Sun[1,2], Dustin R Rubenstein[3], Bo-Fei Chen[1], Shih-Fan Chan[1], Jian-Nan Liu[1], Mark Liu[1], Wenbe Hwang[4], Ping-Shih Yang[2], Sheng-Feng Shen[1]***

[1]Biodiversity Research Center, Academia Sinica, Taipei, Taiwan; [2]Department of Entomology, National Taiwan University, Taipei, Taiwan; [3]Department of Ecology, Evolution and Environmental Biology, Columbia University, New York, United States; [4]Department of Ecoscience and Ecotechnology, National University of Tainan, Tainan, Taiwan

**Abstract** The ability to form cooperative societies may explain why humans and social insects have come to dominate the earth. Here we examine the ecological consequences of cooperation by quantifying the fitness of cooperative (large groups) and non-cooperative (small groups) phenotypes in burying beetles (*Nicrophorus nepalensis*) along an elevational and temperature gradient. We experimentally created large and small groups along the gradient and manipulated interspecific competition with flies by heating carcasses. We show that cooperative groups performed as thermal generalists with similarly high breeding success at all temperatures and elevations, whereas non-cooperative groups performed as thermal specialists with higher breeding success only at intermediate temperatures and elevations. Studying the ecological consequences of cooperation may not only help us to understand why so many species of social insects have conquered the earth, but also to determine how climate change will affect the success of these and other social species, including our own.

**\*For correspondence:** shensf@sinica.edu.tw

**Competing interests:** The authors declare that no competing interests exist.

**Reviewing editor**: Marcel Dicke, Wageningen University, The Netherlands

## Introduction

Social animals, including humans and many insects, have come to dominate the earth, possibly because of their ability to form complex societies (*Laland et al., 2001*; *Fuentes et al., 2010*; *Boyd et al., 2011*; *Wilson, 2012*; *Lucky et al., 2013*). While studies of animal social evolution often emphasize the environment drivers of group-living (*Emlen, 1982*; *Rubenstein and Lovette, 2007*; *Jetz and Rubenstein, 2011*; *Gonzalez et al., 2013*), the ecological consequences of sociality have received less attention. A rare exception comes from our own species, where cooperation is thought to have played a critical role in allowing modern humans to expand rapidly across the earth to exploit a more diverse range of environments than the African savannas in which our ancestors evolved (*Laland et al., 2001*). This shift from being a habitat specialist to generalist, and the subsequent ecological dominance by social species, has been termed the social conquest hypothesis (*Wilson, 2012*). Although this idea has drawn attention from a variety of disciplines, it has proven difficult to test empirically (*Richerson and Boyd, 2008*; *Fuentes et al., 2010*).

Animals derive a variety of cooperative benefits from living in groups (*Alexander, 1974*; *Shen et al., 2014*). Identifying the specific type of benefit individuals receive may help determine the ecological consequences of sociality. If the primary benefit of grouping is to cope with environmental challenges (e.g., predation risk, fluctuating climates, or interspecific competition) (*Alexander, 1974*; *Korb and Foster, 2010*; *Jetz and Rubenstein, 2011*; *Celiker and Gore, 2012*; *Shen et al., 2012*; *Gonzalez et al., 2013*), cooperation should translate into individuals adopting a generalist strategy that allows them to live in a broad range of conditions and cope with a variety of environmental challenges.

**eLife digest** The ability to live and work together in groups likely helped the earliest humans to leave their savannah homes in Africa and successfully settle around the globe. In doing so, humans shifted from being savannah specialists to generalists able to cope with a range of different environments. Cooperation is also believed to be a key to the global success of social insects like bees and ants. However, testing the idea that cooperation allows animals to become generalists that thrive in diverse environments—an idea referred to as the 'social conquest hypothesis'—is difficult.

Climate change has added a new sense of urgency to understanding how species adapt to changing environments, and some studies of humans and other animals have suggested that cooperation may increase or decrease in changing environments. Living in social groups has both benefits and drawbacks: it helps some animals to avoid being eaten by predators, but it also creates more competition for mates, food or other resources. As such, predicting how climate change will impact human and animal societies has also been difficult to test.

Sun et al. have now tested the social conquest hypothesis by looking at how changes in environmental conditions affect the social behavior of the burying beetle. These insects find dead animals and then bury them to be eaten by their larvae. Burying beetles often fight each other to ensure that their own young get exclusive access to a food source. However, working together allows the beetles to bury a carcass before flies and other competitors discover it. Sun et al. compared how much the beetles cooperated at different elevations in the mountains of Taiwan. At each elevation the beetles faced different challenges: higher elevations were colder but had fewer flies, while lower elevations were warmer but had more flies.

Although burying beetles tended to work together more at warmer elevations, where the competition from flies was the most intense, beetles that cooperated with each other were able to successfully breed at all elevations. On the other hand, beetles that were less cooperative were best adapted to raising their young at more moderate elevations, where the climate and competition were less harsh. Similar results were seen when Sun et al. created non-cooperative and cooperative groups of beetles at different elevations and provided each group with a rat carcass. Further experiments that used heaters to artificially warm the carcasses directly proved that cooperation among beetles was indeed encouraged by higher temperatures.

Many studies have suggested that global warming might cause higher levels of conflict in human societies. But by studying how changes in an environment impact cooperation in burying beetles, Sun et al. provide new insights into how climate change may affect the future success of other social animals, including humans.

In contrast, when species form groups as an adaptation to intraspecific challenges (e.g., competition with conspecific groups or with members of their own group over a lack of breeding vacancies or critical resources; *Emlen, 1982*; *West et al., 2006*; *Reeve and Hölldobler, 2007*; *Gonzalez et al., 2013*; *Hsiang et al., 2013*), cooperation should enable individuals to specialize in a single environment (*Figure 1*).

The contrast between habitat specialist and generalist strategies derives from ecological niche theory (*Levins, 1968*; *Futuyma and Moreno, 1988*). Although niche theory has been used to investigate a range of ecological phenomena including species interactions (*Kassen, 2002*), geographic distributions (*Peterson et al., 2011*) and the ecological consequences of climate change (*Clavel et al., 2010*), to our knowledge it has not yet been applied to social evolution. To understand how sociality influences niche breadth evolution, social and non-social populations from the same species need to be examined in a variety of different environments. That is, rather than focusing upon the ecological interactions of a species as a whole, one could separate a species' total niche into different phenotypic components and then determine how these phenotypes influence fitness in varying environments (*Roughgarden, 1972*; *Bolnick et al., 2010*). For social species, total niche breadth can be partitioned into the 'cooperative' and 'non-cooperative' phenotypes, which correspond to generalist and specialist strategies, respectively if the grouping benefit is to cope with harsh environments or severe interspecific competition.

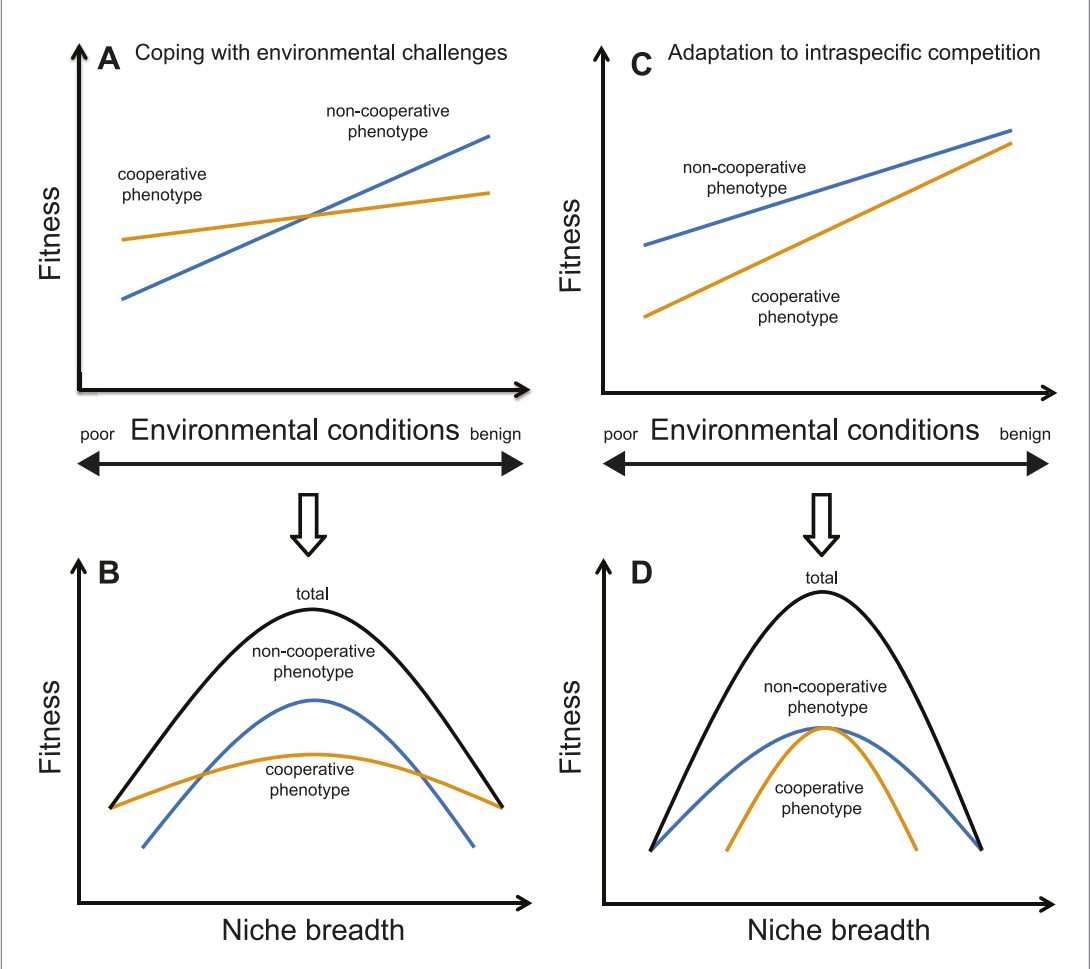

**Figure 1**. Illustration of two different causes of sociality, and their ecological consequences (i.e., niche breadth). (**A**) If cooperation is for coping with harsh environments or interspecific competition, cooperative phenotypes (i.e., forming groups; orange lines) will have higher fitness than non-cooperative phenotypes in poor environments or when the pressure of interspecific competition is high. However, non-cooperative phenotypes (i.e., being solitary; blue lines) could have higher fitness in favorable environments because there are few benefits of cooperating. (**B**) Under such a scenario, a species' total niche breadth (black lines) is expanded due to the cooperative phenotype because a social species' total niche breadth equals to the sum of the cooperative and non-cooperative phenotypes. (**C**) In contrast, if cooperation is the best-of-a-bad-job strategy as a response to intraspecific challenges, the per capita reproductive success will be lower in groups than solitary pairs. This scenario often occurs when grouping occurs because of a lack of critical resources, such as when breeding territories are limited in many cooperatively breeding birds (***Emlen, 1982***). Therefore, cooperative phenotypes do not necessarily have higher fitness than non-cooperative phenotypes in either poor or favorable environments. (**D**) As a consequence, cooperative phenotypes will have little influence on the total niche breadth of a species when cooperation is a response to intraspecific challenges. Note that the trade-offs between specialist and generalist strategies occur only in the case of coping with environmental challenges or interspecific competition, and not in the case of adaptation to intraspecific competition.

Here we examine how group-living impacts the generalist-specialist behavioral tradeoff and its subsequent effect on niche breadth (defined as a thermal performance that influences elevational distribution) in the facultative cooperatively breeding burying beetle (*Nicrophorus nepalensis*). The primary benefit of cooperative breeding behavior in burying beetles is to jointly prepare and bury carcasses more rapidly than their primary competitor, carrion-feeding flies (***Table 1***) (***Eggert and Müller, 1992***; ***Scott, 1994***; ***Trumbo, 1995***). We consider how intraspecific cooperation drives the evolution of thermal specialist vs generalist strategies along an elevational gradient where the degree of temperature-mediated interspecific competition with flies for resources (carcasses) varies with elevation. To determine how temperature influences the degree of interspecific competition, which in turn mediates the cooperative and competitive strategies of *N. nepalensis*, we first documented the

**Table 1.** Identification and abundance of carrion-feeding insects collected on rat carcasses from June to August 2011.

| Order | Percentage (%) | Family | Frequency |
|---|---|---|---|
| Coleoptera | 6.18 | Hydraenidae | 6 |
| | | Leiodidae | 11 |
| | | Ptiliidae | 6 |
| | | Silphidae | 9 |
| Diptera | 91.89 | Anthomyiidae | 7 |
| | | Calliphoridae | 117 |
| | | Carnidae | 2 |
| | | Drosophilidae | 33 |
| | | Fanniidae | 67 |
| | | Muscidae | 63 |
| | | Mycetophilidae | 1 |
| | | Phoridae | 103 |
| | | Psychodidae | 11 |
| | | Sarcophagidae | 7 |
| | | Sciaridae | 5 |
| | | Sphaeroceridae | 60 |
| Hymenoptera | 1.74 | Formicidae | 8 |
| | | Vespidae | 1 |
| Lepidoptera | 0.19 | Tortricidae | 1 |
| Total | 100 | 19 families | 518 |

natural patterns of group size, cooperation, breeding success, and the degree of interspecific competition with flies along the elevational gradient. We then experimentally manipulated the group size of *N. nepalensis* and the degree of interspecific competition with flies to determine the mechanisms underlying the fitness patterns along the elevational gradient.

## Results and discussion

We began by quantifying the natural patterns of group size, cooperative behavior, and breeding success along an elevational gradient in central Taiwan (*Figure 2*) where daily minimum air temperature decreased with increasing elevation ($\chi^2_1 = 222.50$, p<0.001, *n* = 116). We found that group size decreased with increasing elevation (*Figure 3A*) and decreasing air temperature (*Figure 3B*). Furthermore, the probability of breeding successfully varied unimodally along the elevational (*Figure 3C*) and air temperature gradients (*Figure 3D*), peaking at intermediate elevations and air temperatures. Additionally, cooperative behavior—quantified as levels of cooperative carcass processing ('Materials and methods')—increased with increasing group size (*Figure 4*), suggesting that the greater breeding success at higher elevations was due to the cooperative behavior of groups.

To further determine how cooperation influences breeding success in different environments, we created small, non-cooperative groups (one male and one female, *n* = 53) and large, cooperative groups (three males and three females, *n* = 39) at 23 sites along the elevational gradient by placing locally trapped beetles on rat carcasses in specially designed breeding chambers that allowed flies and other small insects to move in-and-out of the chambers freely, but that limited the natural access of beetles (*Figure 5*). Initial group size simulated the number of beetles attracted to odorants produced by decomposing vertebrate carcasses, and the timing of beetle placement mimicked the natural pattern of arrival times, which are longer at higher elevations. We found that the probability of breeding successfully for small and large groups varied along the elevational gradient such that large groups performed as thermal generalists with similar breeding success at all elevations (*Figure 6A*) and air temperatures (*Figure 6B*), whereas small groups performed as thermal specialists with high breeding

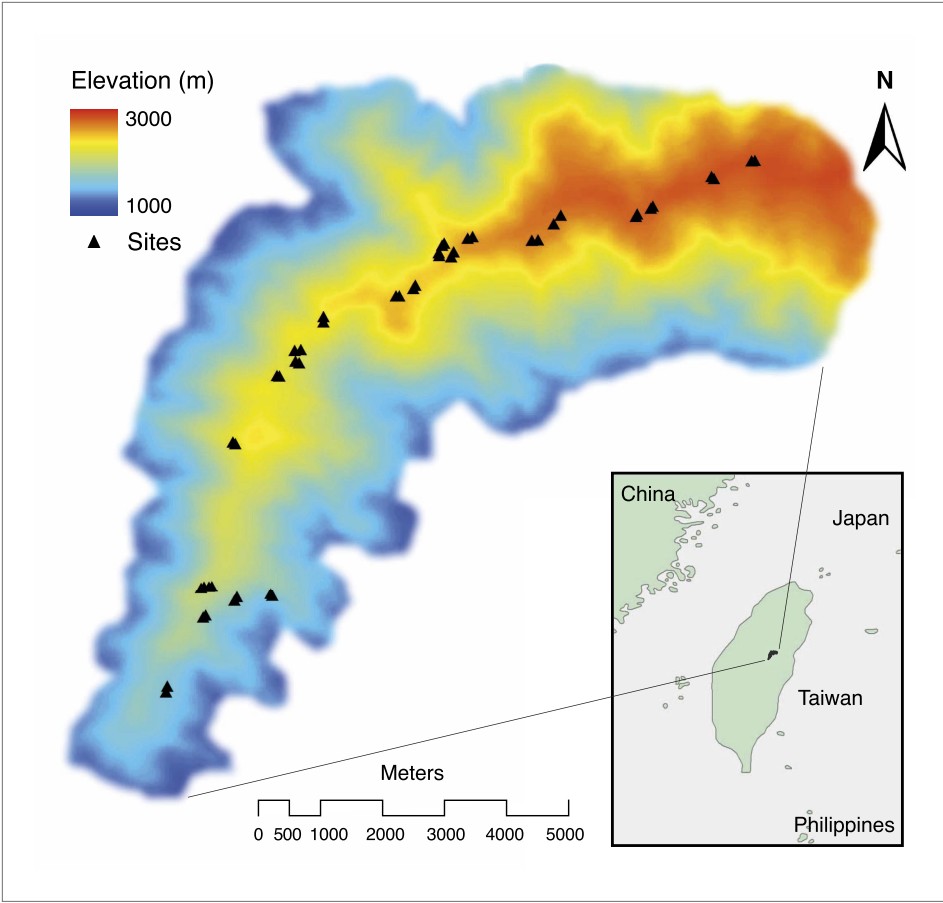

**Figure 2**. Spatial distribution of study sites (black triangles) along an elevational gradient in Nantou, Taiwan (24°5' N, 121°10' E).

success only at intermediate elevations (*Figure 6A*) and air temperatures (*Figure 6B*). Moreover, large groups had higher breeding success than small groups at low elevations (*Figure 6A*) and at warmer temperatures (*Figure 6B*), but small groups had marginally higher breeding success than large groups at intermediate elevations (*Figure 6A*) and temperatures (*Figure 6B*). There were no significant differences in breeding success between large and small groups at high elevations (*Figure 6A*) and low air temperatures (*Figure 6B*).

To establish why breeding success varied with elevation and temperature differently in cooperative and non-cooperative groups, we quantified levels of cooperative carcass processing in our group size treatments across the elevational gradient. We found no relationship between cooperative carcass processing and elevation (*Figure 7A*) or air temperature (*Figure 7B*) in small groups. However, investment in cooperative carcass processing in large groups increased with decreasing elevation (*Figure 7A*) and increasing air temperature (*Figure 7B*), presumably because carcasses decompose more quickly at lower elevations (*Figure 8A*) where fly abundance (*Figure 8B*) and activity (*Figure 8C*) is highest. Experimental exclusion of flies from carcasses confirmed that flies indeed enhance carcass decomposition rates; the mean dry weight of carcasses from which flies were excluded was more than two times heavier than carcasses for which flies had access (*Figure 9*). Our data further showed that in large groups, per capita social conflict ('Materials and methods') varied unimodally with a peak at intermediate elevations and air temperatures (*Figure 10*). Importantly, only investment in cooperative carcass processing, and not social conflict, increased with increasing temperature in large groups. Together these results indicate that an individual's cooperative and competitive strategies are not influenced directly by temperature-dependent physiological constraints per se because higher ambient temperatures typically reduce the cost of activity for ectotherms (*Angilletta, 2009*).

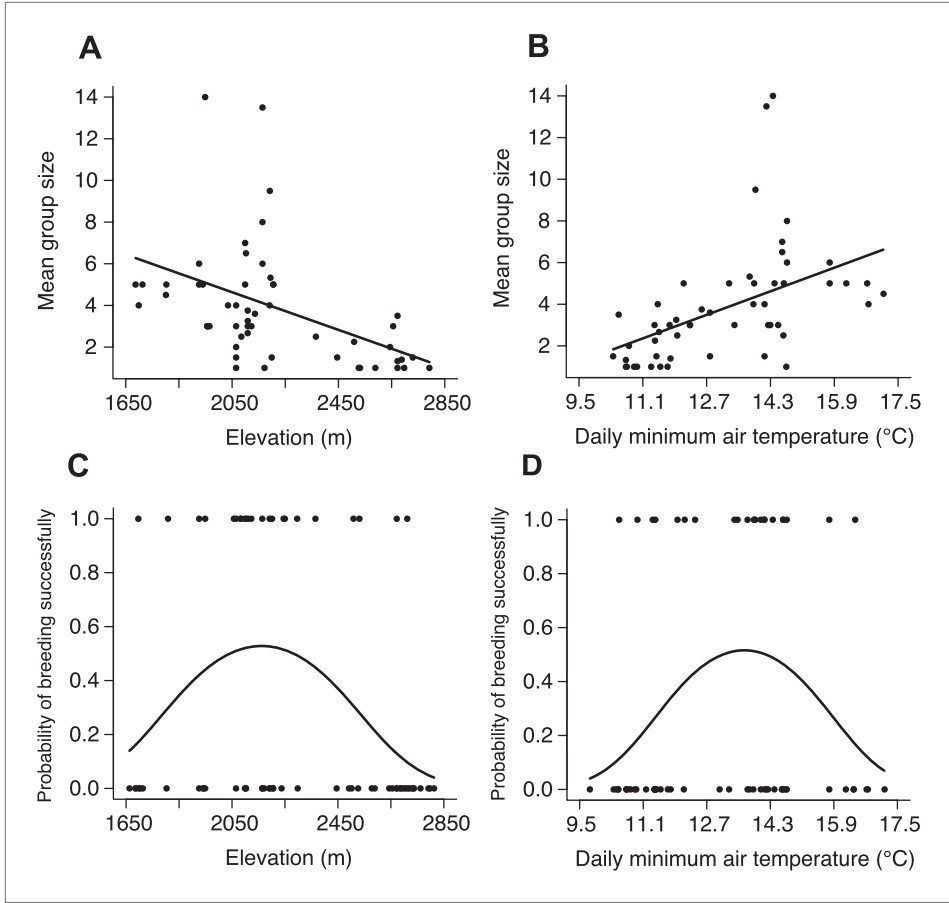

**Figure 3**. Natural patterns of group size and the probability of breeding successfully in relation to elevation and daily minimum air temperature. Mean group size in natural populations decreased with (**A**) increasing elevation ($\chi^2_1$ = 16.26, p<0.001, n = 54) and (**B**) daily minimum air temperature ($\chi^2_1$ = 15.26, p<0.001, n = 53). The probability of breeding successfully in natural populations varied unimodally along (**C**) the elevational ($\chi^2_2$ = 8.68, p=0.013, n = 70) and (**D**) daily minimum air temperature gradients ($\chi^2_2$ = 6.37, p=0.041, n = 66).

Instead, our experiments suggest that an individual's cooperative and competitive strategies are influenced by temperature-mediated interspecific competition for resources, which increases with increasing temperature.

Experimental exclusion of flies from carcasses confirmed that interspecific competition between beetles and flies reduces beetle breeding success; the probability of beetles breeding successfully in small groups along the elevational gradient (from 1664 m to 2809 m) was lower when flies had access to carcasses than when they were excluded (***Figure 11***; for additional details see fly competition treatment in 'Materials and methods'). To determine if temperature mediates this competition, we simultaneously manipulated group size and the degree of competition with flies along the portion of the elevational range where small groups had higher breeding success. We found that experimentally heating carcasses ('Materials and methods') increased fly abundance (***Figure 8B***) and activity (***Figure 8C***) relative to controls. If temperature-mediated competition with flies at low elevations explains why large groups had higher breeding success than small groups, then our heated carcass treatment at higher elevations should have decreased the probability of breeding successfully in small but not large groups. In support of this prediction, we found that heating carcasses differentially affected the breeding success of small and large groups when controlling for elevation such that the probability of breeding successfully in small groups decreased in the heated carcass treatments (***Figure 12A***), but the probability of breeding successfully for large groups remained the same (***Figure 12A***). Moreover, individuals were more cooperative in carcass processing in the heated carcass treatments than in the controls (***Figure 12B***).

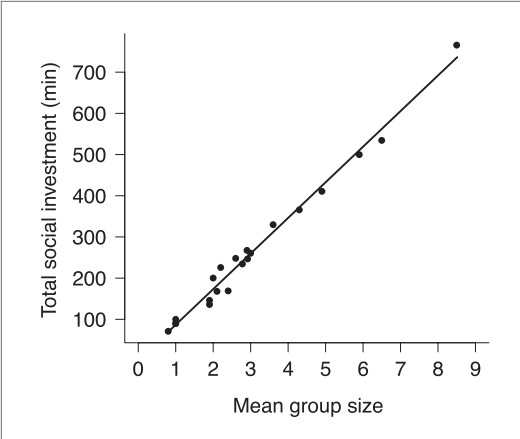

**Figure 4**. The relationship between group size and total investment in cooperative carcass processing in natural groups. Total social investment (minutes, min) in cooperative carcass processing increased with the increasing group size ($\chi^2_1$ = 1681.10, p<0.001, n = 21).

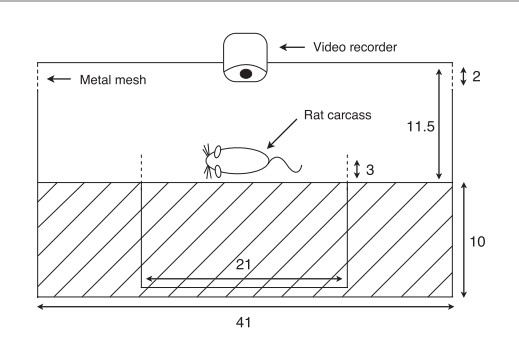

**Figure 5**. Diagram of the experimental container. The apparatus consisted of a larger plastic container to isolate the carcass from scavengers, but beetles and flies were allowed to move freely between the chamber and the outside environment. A smaller container with a rat carcass was provided for burial in the center of the larger container. The entire burial process and behavioral assays were recorded with a video-recorder. Dashed lines represent places connected to long pipes, which allowed beetles to leave the box. Cross hatching indicates the soil layer inside the chamber (unit: cm). Our manipulation successfully created different mean group sizes even after some free-living beetles entered the chambers and some experimental beetles left ($t_{58}$ = 15.08, p<0.001).

In summary, our experiments demonstrated that cooperative beetle groups performed as thermal generalists, but non-cooperative groups performed as thermal specialists. This generalist-specialist behavioral tradeoff along the elevational gradient in *N. nepalensis* is generated by the tension between an individual's share of the grouping benefit and the group's productivity. At low elevations where the pressure of interspecific competition with flies is highest, individuals in large groups were not only more cooperative at handling carcasses, but they also engaged in lower levels of social conflict, both of which enabled them to outcompete flies. As a consequence, cooperation enables burying beetles to expand their thermal niche to a warmer region where competitors are more abundant. In contrast, we found that the 'tragedy of the commons' (*Hardin, 1968*; *Rankin et al., 2007*)—that is the degree of social conflict was higher in large groups, which led to a reduction in breeding success relative to small groups—occurred at intermediate elevations where the pressure of interspecific competition with flies was lower. At these intermediate elevations, non-cooperative groups have marginally higher breeding success than cooperative groups because intraspecific social conflict increased in the absence of interspecific conflict. Nonetheless, this within-group conflict has relatively little influence on the ecological dominance of burying beetles because breeding success is still relatively high in large groups in favorable environments, compared with those at elevations where environments are less favorable. We found a similar pattern in the natural populations (i.e., those without group size manipulations) where breeding success was highest at intermediate elevations even though there are many naturally occurring large groups in this region.

This study provides the first experimental evidence consistent with the social conquest hypothesis, which argues that cooperation promotes the evolution of generalist strategies when the primary benefit of living in groups is to cope with environmental challenges, including climate-mediated interspecific competition (*Wilson, 2012*). Preliminary support for this hypothesis comes from a recent comparative study of sponge-dwelling snapping shrimp (*Synalpheus* spp.), showing that eusocial species were more abundant and occupied a broader range of host sponge species than non-social sister species (*Duffy and Macdonald, 2010*). We have shown experimentally in burying beetles that cooperative groups performed as thermal generalists, but non-cooperative groups performed as thermal specialists. Being cooperative enables burying beetles to extend their range to lower elevations where temperatures are warmer and where competitors are more abundant because individuals in large

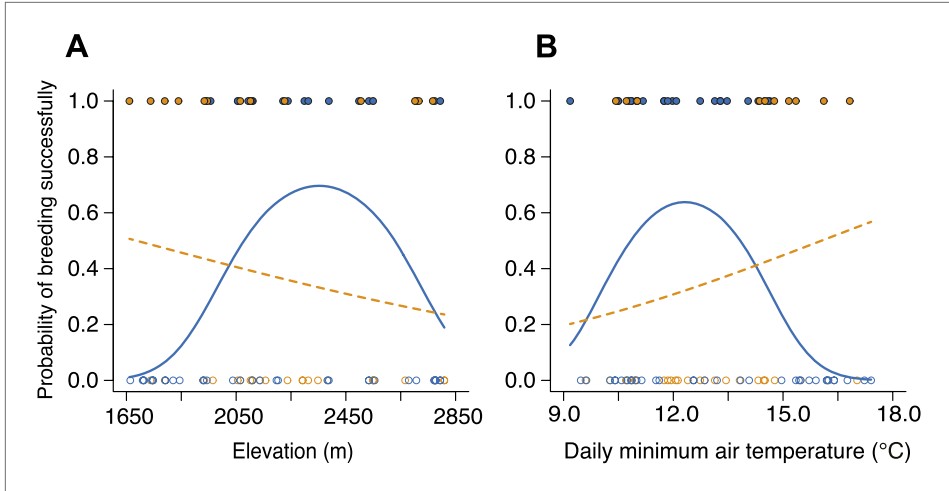

**Figure 6**. Reproductive success varied with group size along elevational and temperature gradients. (**A**) The probability of breeding successfully for small (blue circles, solid line) and large groups (orange circles, dashed line) varied differently along the elevational gradient (group size × elevation interaction, $\chi^2_2$ = 10.56, p=0.005, $n$ = 92; for large groups, $\chi^2_2$ = 3.19, p=0.20, $n$ = 39; for small groups, $\chi^2_2$ = 7.66, p=0.022, $n$ = 53), with large groups having higher breeding success than small groups at lower elevations ($\chi^2_1$ = 5.60, p=0.018, $n$ = 26), but small groups having marginally higher breeding success than larger groups at intermediate elevations ($\chi^2_1$ = 3.51, p=0.061, $n$ = 53). There was no significant difference in breeding success between small and larger groups at high elevations ($\chi^2_1$ = 0.04, p=0.84, $n$ = 13). (**B**) The probability of breeding successfully for small and large groups also varied differently along the daily minimum air temperature gradient (group size × temperature interaction, $\chi^2_2$ = 7.28, p=0.026, $n$ = 92; for large groups, $\chi^2_2$ = 1.55, p=0.46, $n$ = 39; for small groups, $\chi^2_2$ = 6.15, p=0.046, $n$ = 53), with large groups showing higher breeding success than small groups at higher temperatures ($\chi^2_1$ = 5.60, p=0.018, $n$ = 26), but small groups having marginally higher breeding success than small groups at intermediate temperatures ($\chi^2_1$ = 3.46, p=0.063, $n$ = 53). Again, there was no significance in breeding success between small and larger groups at high temperatures ($\chi^2_1$ = 0.0001, p=0.99, $n$ = 13). Open circles indicate failed breeding attempts and closed circles indicate successful breeding events. Solid lines denote predicted relationships from GLMMs, whereas dashed lines denote statistically non-significant relationships.

groups were more cooperative at handling carcasses, which enabled them to outcompete flies. Thus, cooperation allows burying beetles to expand their thermal niche into an environment from which they would otherwise be competitively excluded. Ultimately, studying the ecological consequences of cooperation may not only help us to understand why so many species of social insects have conquered the earth, but also to determine how climate change will affect the success of these and other social species, including our own.

## Materials and methods

### Study area
The elevational gradient in central Taiwan (*Figure 2*) covers broadleaf forest at lower elevations and mixed conifer-broadleaf forest at higher elevations. We chose study sites primarily in mature forests and avoided cultivated or open areas.

### Group size in natural populations
We conducted a preliminary investigation of the natural pattern of arrival times of free-ranging beetles on carcasses along the elevational gradient from August to September 2012 and from June to September 2013. In each trial, a 75 g rat carcass was presented on the soil and covered with a 21 × 21 × 21 cm iron cage with mesh size of 2 × 2 cm to prevent vertebrate scavengers. We video recorded the entire burial process. Because video recordings showed that the number of beetles on the carcass varied with time, we determined the mean group size (an average group size of the maximum number of beetles sampled every hour) before the burial was complete. Beetle arrival time was

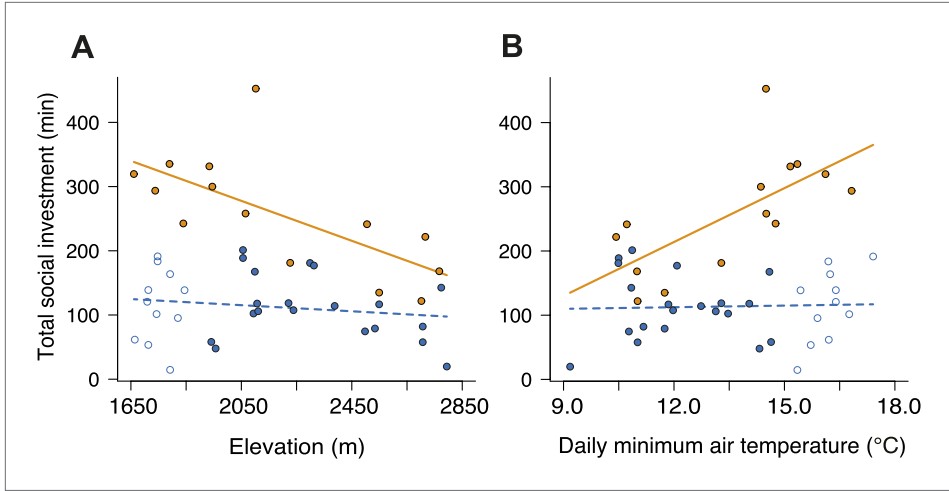

**Figure 7**. Investment in cooperative carcass processing along the elevational and temperature gradients. Investment (minutes, min) in large (closed orange circles, solid line) and small groups (closed blue circles, successful trials; open blue circles, failed trials; dashed line) varied along the (**A**) elevational (group size × elevation interaction, $\chi^2_1 = 7.65$, p=0.006, n = 45) and (**B**) daily minimum air temperature gradients (group size × temperature interaction, $\chi^2_1 = 9.90$, p=0.002, n = 45) such that investment in large groups decreased with (**A**) increasing elevation ($\chi^2_1 = 10.30$, p=0.001, n = 14) and (**B**) decreasing daily minimum temperature ($\chi^2_1 = 9.93$, p=0.002, n = 14). There was no relationship between cooperative carcass processing and (**A**) elevation ($\chi^2_1 = 0.80$, p=0.37, n = 31) or (**B**) daily minimum air temperature ($\chi^2_1 = 0.04$, p=0.84, n = 31) in small groups. Solid lines denote predicted relationships from GLMs, whereas dashed lines denote statistically non-significant relationships.

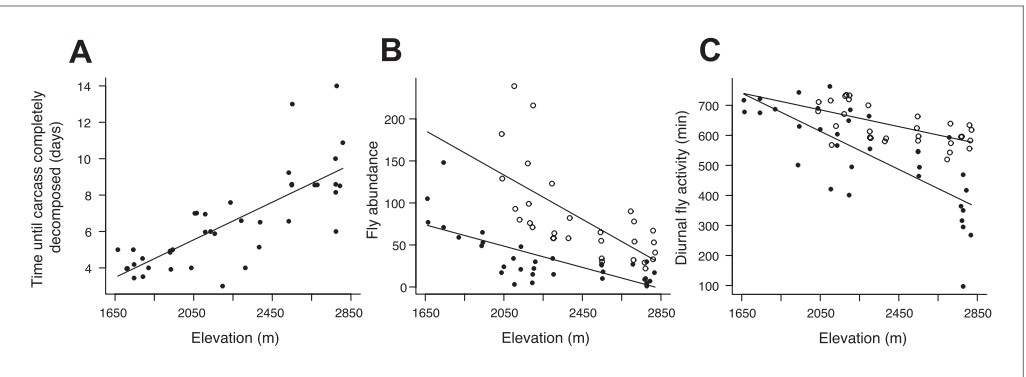

**Figure 8**. Carcass decomposition rates and the effect of experimentally heating carcasses on fly abundance and activity along the elevational gradient. (**A**) The time until the carcass was completely consumed by carrion-feeding insects increased with increasing elevation ($\chi^2_1 = 50.87$, p<0.001, n = 40). The control treatments (closed circles) represent the natural patterns of fly abundance and activity on carcasses. (**B**) Fly abundance decreased with increasing elevation ($\chi^2_1 = 21.49$, p<0.001, n = 33), but heated carcass treatments (open circles) showed higher fly abundance than controls (closed circles) ($\chi^2_1 = 42.65$, p<0.001, n = 55). (**C**) Diurnal fly activity decreased with increasing elevation ($\chi^2_1 = 39.90$, p<0.001, n = 33), but flies were more active on heated carcass treatments than on controls ($\chi^2_1 = 29.85$, p<0.001, n = 55). Solid lines denote predicted relationships from GLMs.

determined when the first burying beetle was observed on the carcass. The arrival time of free-ranging burying beetles on carcasses increased with increasing elevation ($\chi^2_1 = 24.41$, p<0.001, n = 73).

## Heterotrophic succession and fly competition

To confirm that flies (Diptera) are the major competitors of burying beetles, we first examined the succession pattern of carrion-feeding insects on 150 g (n = 5) and 200 g (n = 7) rat carcasses. This experiment was conducted at an intermediate elevation (2000 m) from June to August 2011. Initially, rat

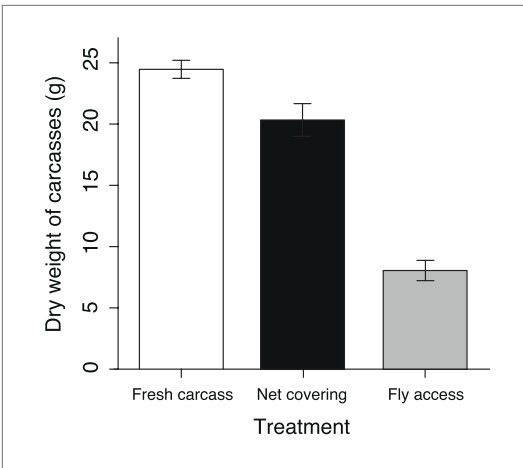

**Figure 9**. Remaining carcasses dry weight after exposure in different treatments. The mean ± SE remaining carcass dry weights in the fresh carcass controls (white column, n = 9) were significantly heavier than those in the net covering (black column, n = 9) and fly access treatments (grey column, n = 9) ($\chi^2_2$ = 145.66, p<0.001, n = 27).

carcasses were placed at 50 m intervals along the ground and covered by 21 × 21 × 21 cm iron cages following the previous procedure. Samples were collected daily in the morning (between 10:00 and 12:00) for three days to resemble the insect community at an early successional stage. Mean abundances of carrion-feeding insects on 12 carcasses were examined daily after exposure, continuing for 1 day (n = 5), 2 days (n = 3), and 3 days (n = 4). For each sampling period, we first used an aerial sweep net to collect flying insects before the carcass was moved. We then collected all insects present on the carcass. Finally, the soil beneath each carcass was sampled within a sieve tray (2500 cc), and insects were extracted by a modified Berlese funnel (**Newell, 1955**). All specimens were preserved in 70% ethanol for further identification in the laboratory. Taxonomic determination was made to the family level (**Borror et al., 1989**).

In total, 518 adult carrion-feeding insects were collected, representing 29 families in four orders (**Table 1**), including necrophagous, saprophagous, and omnivorous species (**Smith, 1986**). Of these, Diptera and Coleoptera were the two most represented groups, constituting 98.1% of the individuals captured. A GLM was performed to assess if the abundance (number of individuals per carcass) differed between insect families (Diptera and Coleoptera) using carcass weight and the day after carcass exposure as covariates.

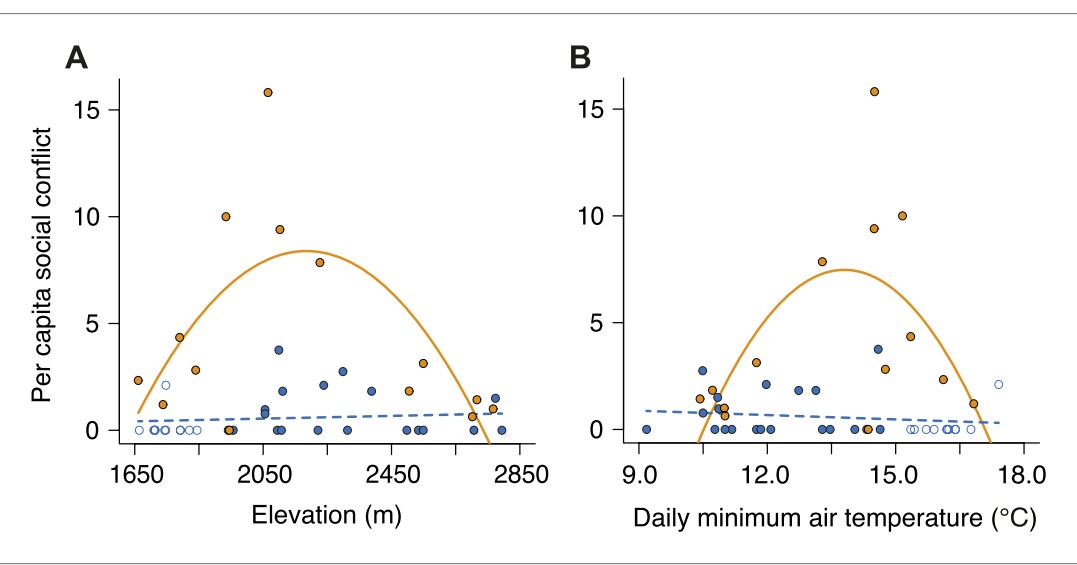

**Figure 10**. Per capita social conflict in small and large groups along the elevational and temperature gradients. Patterns of per capita social conflict differed between small (closed blue circles, successful trials; open blue circles, failed trials; dashed line) and large groups (closed orange circles, solid line) along gradients of (**A**) elevation (group size × elevation interaction, $\chi^2_2$ = 14.73, p<0.001, n = 45) and (**B**) daily minimum air temperature (group size × temperature interaction, $\chi^2_2$ = 13.98, p<0.001, n = 45). In large groups, per capita social conflict varied unimodally with elevation ($\chi^2_2$ = 9.11, p=0.011, n = 14) and daily minimum air temperature ($\chi^2_2$ = 6.17, p=0.046, n = 14), peaking at intermediate elevations and temperatures. However, in small groups, per capital social conflict did not vary with elevation ($\chi^2_2$ = 4.37, p=0.11, n = 31) or daily minimum air temperature ($\chi^2_2$ = 0.73, p=0.70, n = 31).

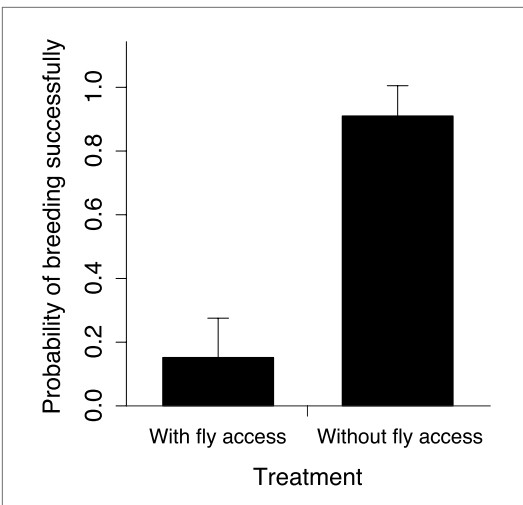

**Figure 11**. The probability of breeding successfully in relation to fly accessibility. Mean ± SE probability of breeding successfully (GLMM fitted values) in small groups was higher when flies were excluded from carcasses ($n$ = 18) than when they had access to carcasses ($n$ = 23) along the elevational gradient ($\chi^2_1$ = 12.06, p<0.001, $n$ = 41).

We found that the mean abundance of Diptera was significantly higher than that of Coleoptera ($\chi^2_1$ = 49.85, p<0.001, $n$ = 12).

## Preparation of animals

Burying beetles were collected by hanging pitfall traps baited with 100 ± 10 g of rotting chicken. Pitfall traps were checked each morning. Beetles were housed individually in 320 ml transparent plastic cups and fed with mealworms (*Zophobas morio*) if they were kept more than three days before the experiment. Each beetle was weighed to the nearest 0.1 mg and marked with Testors enamel paint on the elytra (*Butler et al., 2012*) for individual identification the night before use. Sex was determined by the markings on the clypeus; males have a rectangular, orange marking, whereas females do not.

## Experimental design and procedure

Our experimental chambers consisted of a smaller plastic container (21 × 13 × 13 cm with 10 cm of soil) located inside a larger container (41 × 31 × 21.5 cm with 11 cm of soil) (*Figure 5*). Multiple holes on the side walls of the smaller container permitted beetle movement between the two containers. The cap of the larger container was fitted with a digital camera and was raised up 2 cm by iron mesh to allow entry by free-ranging flies and beetles, but not by vertebrate scavengers (*Figure 5*). Digital cameras were powered by Yuasa lead-acid

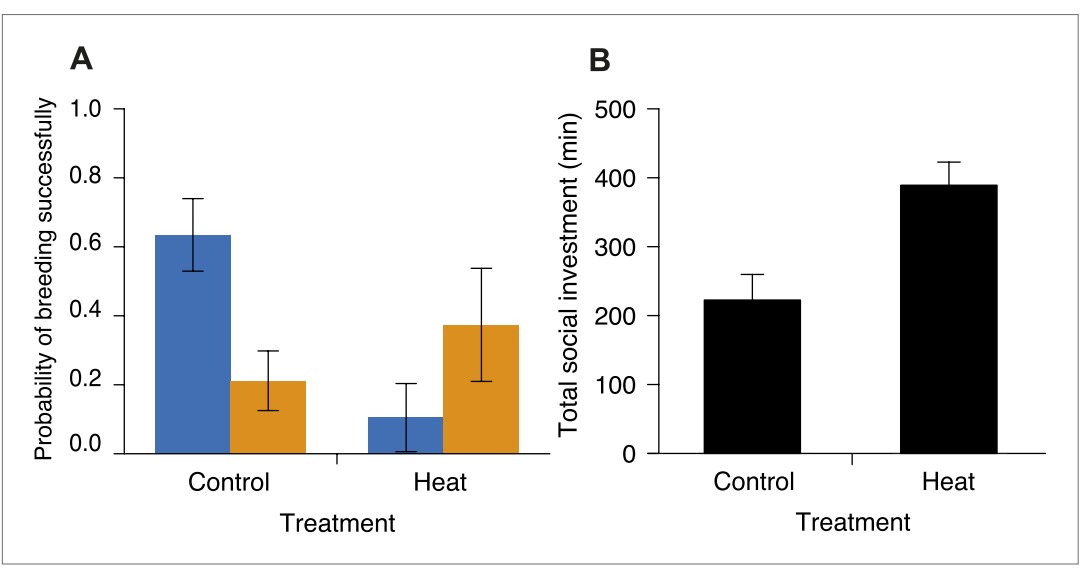

**Figure 12**. Investment in cooperative carcass processing in control and heated carcass treatments along the elevational gradient. Heating carcasses differentially affected the breeding success of small and large groups when controlling for elevation ($\chi^2_1$ = 6.55, p=0.010, $n$ = 116). (**A**) Mean ± SE probability of breeding successfully (GLMM fitted values) for large (orange columns) and small (blue columns) groups of burying beetles in control and heated carcass treatments. Heating carcasses reduced the probability of breeding successfully in small groups ($\chi^2_1$ = 5.99, p=0.014, $n$ = 68), but not in large groups ($\chi^2_1$ = 0.98, p=0.32, $n$ = 48). (**B**) Mean ± SE total investment (minutes, min) in cooperative carcass processing was higher in heated carcass than control treatments ($\chi^2_1$ = 12.67, p<0.001, $n$ = 16).

batteries (6V 12Ah), which were replaced every morning. We measured air temperature every 30 min for the duration of the experiment using Maxim's iButton devices that were placed within the larger container. Based upon the natural pattern of arrival times from our pilot study (see 'Group size in natural populations' in 'Materials and methods'), we released the marked beetles into the experimental apparatus 1 day, 2 days, and 3 days after the trials began at elevations of 1700–2000 m (low), 2000–2400 m (intermediate) and 2400–2800 m (high), respectively.

To quantify breeding success, we exhumed the carcasses approximately 14 days after they were buried and collected third instar larvae, if there were any. Across the 92 trials that were completed successfully, 52 trials resulted in successful breeding attempts and 40 trials contained carcasses that were completely consumed by maggots. The 40 failed trials were used to examine the carcass consumption rate by maggots as an indicator of interspecific competition along the elevational gradient (*Figure 8A*).

## Fly competition treatment

To assess the effect of fly competition on carcass decomposition rates, we evaluated the difference in carcass weight loss among net-covered treatments (i.e., fly access was restricted from the entire cage), natural fly access treatments, and fresh carcass controls at intermediate elevations (2100 m). The carcasses of natural fly access treatments were exposed to flies until maggots finished consuming and left the carcasses. The dried weights of all carcasses were obtained by dehydrating the carcasses to a constant weight in a drying oven at 65°C. We also compared the probability of breeding successfully in treatments where flies had access to the carcasses and those where flies were excluded along the elevational gradient (from 1664 m to 2809 m).

## Carcass heating treatment

To explore temperature-mediated cooperation in response to fly competition in situ, a heating device was continuously applied underneath each carcass to provide a warming effect. To determine if heating carcasses made them more attractive to flies, we compared fly activity and abundance on heated carcasses to those of control treatments on the day we released the beetles in each trial. Fly activity was quantified as the total duration between the first fly arriving at the carcass and the last fly leaving the carcass, whereas fly abundance was quantified as the total number of flies video recorded between 6:00 to 18:00 at 30-min intervals. The heating device was constructed with a series circuit of cement resistors (40 Ω), which was powered by Yuasa lead-acid batteries (6V 12Ah). The soil temperature differences between the heated carcass treatment and its ambient environment were measured using thermal probes at a depth of 5 cm daily in the morning in 32 trials. On average, the heated carcass treatment created higher soil temperatures (28.7 ± 0.71°C) than those of ambient environment (17.4 ± 0.31°C) ($\chi^2_1 = 212.06$, p<0.001, $n = 64$). Further, a total of 24 heated carcass treatments were conducted along the elevational gradient (from 2039 m to 2814 m) where small, non-cooperative groups had higher breeding success.

## Behavioral assays

In total, 4488 hr of video were recorded from the 92 successful non-heated (control) trials ($n = 39$ large groups, 53 small groups) and 1170 hr from the heated carcass treatments (n = 9 large groups, 15 small groups). A variety of social behaviors, including per capita social conflict and investment in cooperative carcass processing, were scored on the first night (from 19:00 to 05:00) using the Observer Video-Pro software (Noldus) for the 34 successful breeding trials ($n = 14$ large groups, 20 small groups) and 11 trials of small groups failed at the lower elevations (from 1664 m to 1844 m). Aggressive interactions were defined as social conflict if a beetle grasped, bit, chased, or escaped from the other same-sexed individual. A sample video of aggressive interaction can be seen in *Video 1*. We measured per capita social conflict as the total number of aggressive interactions divided by mean group size for each observation period. To

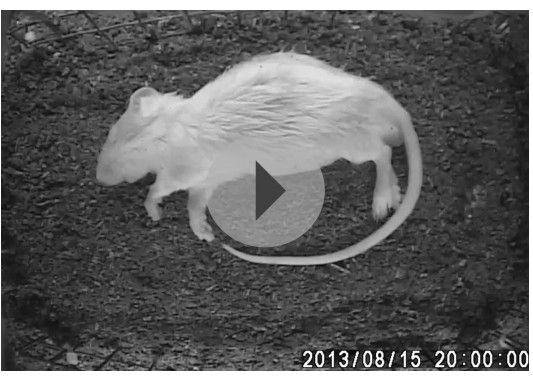

**Video 1**. Social investment, Large group, August 15, 2011.

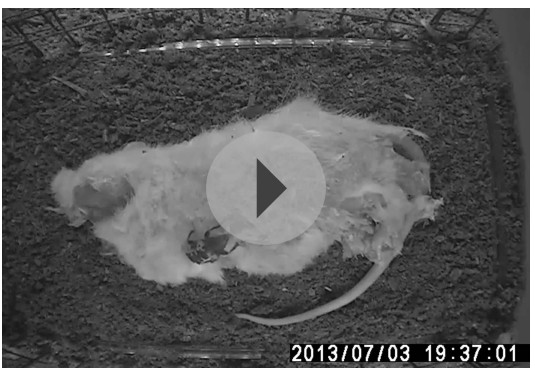

**Video 2**. Social conflict, Large group, July 3 2013.

quantify total social investment in cooperative carcass processing, we estimated the cumulative time that each beetle spent depilating rat hair, removing maggots, or digging soil during carcass burial and preparation. A sample video of cooperative carcass processing can be seen in *Video 2*. Investment was quantified as the duration of cumulative time sampled for a 10 min observation period in each hour (100 min in total).

## Data analysis

Multivariate analyses were performed using generalized linear models (GLMs). If the random effects of repeated sampling of study sites were required, generalized linear mixed models (GLMMs) were used. To test for the differences in the probability of breeding successfully between the two group sizes and carcass heating treatments along the elevational and temperature gradients, the outcome of breeding success (1 = Success, 0 = Failure) was fitted as a binomial response term. The environmental factors (elevation and daily minimum air temperature), group size treatments, and carcass heating treatments were fitted as covariates of interest. For the carcass heating treatments, the fitted value of the probability of breeding successfully was compared between heated carcass and control treatments. All statistic analyses were performed in the R statistical software package (*R Core Team, 2012*).

## Acknowledgements

We acknowledge Yen-Cheng Lin, Tzu-Neng Yuan, Ching-Fu Lin, and Yu-Ching Liu for their great support in the field. We also thank Wei-Ping Chan for help making *Figure 2* and the staff at Mei-Feng Highland Experiment Farm, National Taiwan University for the logistic help.

## Additional information

### Funding

| Funder | Grant reference number | Author |
| --- | --- | --- |
| Academia Sinica | Career Development Award | Sheng-Feng Shen |
| National Science Council of Taiwan | NSC101-2313-B001-008-MY3 | Sheng-Feng Shen |
| National Science Foundation | IOS-1121435 | Dustin R Rubenstein |
| National Science Foundation | IOS-1257530 | Dustin R Rubenstein |

The funders had no role in study design, data collection and interpretation, or the decision to submit the work for publication.

### Author contributions

S-JS, Conception and design, Acquisition of data, Analysis and interpretation of data, Drafting or revising the article; DRR, S-FS, Conception and design, Analysis and interpretation of data, Drafting or revising the article; B-FC, Conception and design, Acquisition of data; S-FC, Acquisition of data, Analysis and interpretation of data; J-NL, ML, Analysis and interpretation of data, Drafting or revising the article; WH, P-SY, Conception and design, Drafting or revising the article

### Ethics

Animal experimentation: All of the animals were handled according to approved Biosafety Committee protocols of the Academia Sinica. The protocol was approved by the Biosafety Committee of Academia Sinica (Permit Number:BSF0412-00002446).

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
