## [Decision Letter]

Thank you for sending your work entitled “Climate-mediated Cooperation Promotes Niche Expansion” for consideration at *eLife*. Your article has been favorably evaluated by a Senior editor and 2 reviewers, one of whom is a member of our Board of Reviewing Editors.

Your study addresses an interesting question, i.e., whether cooperative behaviour allows a species that experiences environmental challenges such as interspecific competition to occupy a wider range of niches than when cooperative behaviour is not exhibited. It includes both natural history elements documenting the actual situation along an elevation (and temperature) gradient followed by directed manipulative experiments. Overall the manuscript is well presented, but at some instances, clarity can be enhanced. The reviewers have also raised some concerns, especially relating to two figures that are central for the main conclusion of the study. In addition there is a concern about the statistical analysis to compare groups. This is explained below. The response to the concerns will be crucial in the decision on your manuscript.

Two figures that are central for the main conclusion: your study looks at group size and reproductive success in burying beetles as a function of a temperature/elevation gradient. Correlation data shows that burying beetle groups are larger at low elevation/high temperature, which may be due to factors like carcasses decomposing more quickly or that there are more beetles at low elevation. In addition you performed impressive field manipulations where you vary the group size (small and large) of semi natural beetle groups and follow how they perform across the gradient. This shows differences between the two groups sizes in success along the gradient.

You interpret this in light of the idea that social life allows organisms to invade new habitats where they would fail without sociality. Overall these are interesting data and an interesting idea but there are some concerns. The manipulation experiment data in Figures 5 and 6 are key to concluding that the large groups are more tolerant to environmental variation than small groups. However, looking at Figure 5, it looks like the large groups are doing worse in the intermediate values (and p=0.06 for the chi squared suggests that this is likely real). But you have drawn a straight line through things for large groups and one reviewer is worried that this is missing an important detail: small groups are best at intermediate temperatures and elevations. Moving to Figure 6, it is argued that there is no trend for the small groups but also, there are no small groups at all at low elevations/high temperature and if one removed the large groups at these points, one would lose the correlation for large groups I expect (was this done?). Without this, it is hard to know whether the correlation is a direct effect or the large groups or just down to the fact that small groups cannot make it for the key ranges where temperature effects would be seen.

To remove the concerns we would like to see more analyses that follow up on the above 1) Are small groups better under some conditions, 2) What does a mean success plot look like for Figure 5 (it looks like large groups plummet in the middle) and do the two mean plots differ in shape between small and large groups? 3) What happens if one compares data over the same temperature/elevation ranges in Figure 6?

Moreover, there is an additional ‘statistical concern’: there are several places in the text where a non-significant result is compared to a significant result and this is used to imply that one is biologically different to the other: e.g., temperature does not affect small groups (non significant) but it does effect large groups (significant). However, this is not the way to analyse this. Rather, these should be tested against each other to compare them. Finally, the practice of drawing a straight line through non-significant results that have high variance intuitively seems off: please reconsider.

It seems that small groups are good for some ranges and large ones for others and the beetles appear to be responding to this. If correct, then this suggests that the ability to be socially *plastic* allows them to tolerate more of a range in climate. This is a bit different to what you are suggesting: the large groups are no longer generalist but the species range is likely increased by being able to make large or small groups when needed. So, dependent on the additional analyses, the narrative of the paper would need to be reworked in response to this.

---

## [Author Response]

*[…] Looking at*
Figure 5*, it looks like the large groups are doing worse in the intermediate values (and p=0.06 for the chi squared suggests that this is likely real). But you have drawn a straight line through things for large groups and one reviewer is worried that this is missing an important detail: small groups are best at intermediate temperatures and elevations*.

This is a very important point, and one that we apparently failed to make clear in the original manuscript. As you suggest, we now also interpret the result more clearly as small groups have marginally higher reproductive success than large groups at intermediate temperatures and elevations. These changes make the results fit even better in our generalist-specialist trade-off framework (see Figure 1). That is, small groups perform as thermal specialists and do best at intermediate temperatures and elevations, whereas as large groups are generalists with wider thermal niches, but lower reproductive success. The detailed discussion of this result is now added in the text and the figure legend of new Figure 6.

*Moving to*
Figure 6*, it is argued that there is no trend for the small groups but also, there are no small groups at all at low elevations/high temperature and if one removed the large groups at these points, one would lose the correlation for large groups I expect (was this done?). Without this, it is hard to know whether the correlation is a direct effect or the large groups or just down to the fact that small groups cannot make it for the key ranges where temperature effects would be seen*.

Thank you for pointing this out. We actually did perform our experiments for both large and small groups at all elevations, but since all small group breeding attempts failed at low elevations, we did not show these data in the original figure. In other words, Figure 6 originally only showed total investment of *successful* breeding events to avoid the potential complications of including failed breeding events. We now realize this may have been misleading, so we have added the investment data of failed breeding events to the figure and analyses. Importantly, adding these points has no effect on the analyses, and the original pattern of investment for small groups remains the same (in new Figure 7).

*To remove the concerns we would like to see more analyses that follow up on the above 1) Are small groups better under some conditions, 2) What does a mean success plot look like for*
Figure 5
*(it looks like large groups plummet in the middle) and do the two mean plots differ in shape between small and*
*large groups?*

As suggested, we have added these new analyses. Small groups perform better at intermediate elevations (see response above for additional details) and we have now used regression to represent the non-significant relationship (the original straight line is the mean probability of breeding successfully).

*What happens if one compares*
*data over the same temperature/elevation ranges in*
Figure 6*?*

As we describe above, we have now added investment data of failed breeding event for small groups so that data for large and small groups cover the same temperature and elevational ranges in the new Figure 7.

*Moreover, there is an additional ‘statistical concern’: there are several places in the text where a non-significant result is compared to a significant result and this is used to imply that one is biologically different to the other: e.g., temperature does not affect small groups (non significant) but it does effect large groups (significant). However, this is not the way to analyze this. Rather, these should be tested against each other to compare them*.

Thank you for pointing out the need to perform multivariate analyses in these instances. We agree that this is a more appropriate way to analyze our data. We have added the analyses for every comparison, as suggested. Importantly, all of the results and conclusions remain the same.